# Surveying Psychological Wellbeing in a Post-Pandemic World: The Role of Family and Social Support for LGBTQ+ and Cisgender Heterosexual Adults in the UK

**DOI:** 10.3390/healthcare12161634

**Published:** 2024-08-16

**Authors:** Katie Stokes, Marie Houghton, Jorge Gato, Fiona Tasker

**Affiliations:** 1School of Psychological Sciences, Birkbeck University of London, London WC1E 7HX, UK; kstokes272@hotmail.com (K.S.); mhough01@mail.bbk.ac.uk (M.H.); 2Faculty of Psychology and Education Sciences, University of Porto, 4099-002 Porto, Portugal; jorgegato@fpce.up.pt

**Keywords:** COVID-19 pandemic, LGBTQ+, psychological wellbeing, depression, anxiety, stress, life satisfaction, social support

## Abstract

Studies have emphasized the importance of social support in mitigating the relationship between psychological distress and mental health effects, with family of origin and family of formation providing key sources of social support over the life course. However, LGBTQ+ people may experience family of origin relationships as a source of distress, while partners and friends may buffer the relationship between minority stress and psychological wellbeing. Through our online survey (March–June 2022), which was conducted when the social restrictions of the COVID-19 pandemic were lifted in the UK, we considered the association between psychological wellbeing and sources of social support by sampling n = 1330 LGBTQ+ and cisgender heterosexual adults. LGBTQ+ adults generally experienced poorer psychological wellbeing outcomes than did cisgender heterosexual people. For LGBTQ+ adults, social support from family of origin, a special person, or friends was not associated with depression, anxiety, or stress levels, but social support from family or a special person was positively associated with higher levels of life satisfaction. Our findings indicate the importance of considering negative as well as positive wellbeing.

## 1. Introduction

In December 2019, a novel coronavirus was identified in the city of Wuhan, China. Three months later, on 11 March 2020, the COVID-19 pandemic was declared [1]. In the ensuing weeks, heads of state around the world began imposing measures designed to curb the spread of disease. Stay-at-home orders in the United Kingdom (UK) were issued on 23 March 2020, and two further lock-downs took place during 2020–2021 [2]. Additional actions were implemented to stem the flow of COVID-19, including UK regional lockdowns and regulations regarding social distancing [3]. Such measures led to individuals spending more time alone and less time with people outside of their household. While deemed necessary to keep communities physically healthy, warnings were sounded about the short- and long-term effects of social restrictions on psychological wellbeing, as prior research investigating the link between disease outbreaks and social isolation found psychological consequences [4]. Studies have shown increases in loneliness, anxiety, depression, and perceived stress during the pandemic [5,6,7,8], as well as general decreases in work and life satisfaction among certain groups [9,10]. A study of over 2000 United States (US) citizens compared levels of serious mental distress and found an eight-fold increase between 2018 and April 2020 [11]. Likewise, mental health problems in the UK have similarly soared [12].

When compared with rates found in general population data, lesbian, gay, bisexual, transgender, queer/questioning, plus (LGBTQ+) individuals have been found to be at increased risk of psychological problems during the COVID-19 pandemic [13,14,15] with specific increases in symptoms of depression, anxiety, and alcohol abuse being documented [16,17,18]. Perhaps this increased vulnerability of LGBTQ+ individuals relative to cisgender heterosexual peers during a time of general population stress such as the pandemic is not surprising, given the higher rates of mental health problems experienced by LGBTQ+ people pre-pandemic. The current study, critically timed at a point when both the health threat of COVID-19 was reduced and social restrictions were lifted in the UK, investigated whether LGBTQ+ deficits in psychological wellbeing were still evident and explored the role that social support disparities played in supporting psychological wellbeing. 

### 1.1. Psychological Wellbeing of LGBTQ+ Individuals 

Sexual and gender minority (SGM) individuals have historically exhibited poor psychological wellbeing [19,20,21,22], with estimated suicide risks up to six times greater than in the general population [23]. A meta-review of 199 papers found a greater prevalence of mental health problems among sexual minority (SM) individuals compared to cisgender heterosexual peers [24]. Separately, in 2018, the National Health Interview Survey found that LGB individuals experienced more than double the amount of serious psychological distress than did heterosexual respondents [25], while several other studies revealed lower life satisfaction levels among SM individuals than those of their peers [26,27,28]. 

A myriad of reasons can be proposed as to why SGM people are more at-risk than others from the psychological impacts of COVID-19. The closure of schools and universities leading to the loss of essential support services such as counselling may have a detrimental impact on LGBTQ+ youth and young adults [15]. Indeed, increased anxiety among SGM students has been reported [14]. Employment status may also have a role to play, with approximately 40% of the LGBTQ+ population in the US working in the service industry [29], a sector that laid-off many staff during the pandemic with a corresponding toll on mental health [30,31,32]. In addition to the differential impacts of school and workplace closures on LGBTQ+ communities, minority stress may play a key role in the inequitable ways in which LGBTQ+ people have experienced the pandemic. Meyer’s minority stress model refers to a framework in which stigma, prejudice, and discrimination lead to a hostile environment of chronic psychosocial stressors unique to SGM people that result in negative health outcomes [33]. Numerous studies have outlined how LGBTQ+ individuals, exposed to rejection and discriminating attitudes, have been affected by minority stress [34,35,36,37]. In recent years, Meyer’s model has been expanded to encompass potential protective factors against minority stress, including social support from family and peers, coping skills, self-acceptance, and positive gender and sexual identity [38]. 

### 1.2. Sociodemographic and Personality Vulnerability to Psychological Wellbeing Problems 

Throughout the pandemic, several factors have been associated with variations in psychological wellbeing. Socio-demographic attributes such as age and gender provide clear examples, with studies pointing to younger age groups and women presenting more commonly with mental health problems [39,40,41,42,43]. Personality traits such as neuroticism and optimism also have been reviewed in the context of the pandemic. Specifically, studies have shown a positive relationship between neuroticism and fear of COVID-19 as well as higher rates of anxiety and depression symptoms [44,45,46]. Other studies have found that individual differences in optimism and pessimism predict wellbeing, with pessimism associating with increases in symptoms of depression, anxiety, and fear of COVID-19 [47], while higher levels of optimism predicted lower levels of fear of COVID-19 [48]. Not surprisingly, having a pre-existing physical or mental health problem pre-pandemic was found to worsen psychological wellbeing outcomes [7,49,50,51]. 

### 1.3. Social Support

The extent of social support has been identified as a key protective factor in averting various psychological disorders [52,53]. Perceived social support, which is the degree to which an individual believes themselves to be accepted, loved, and provided with material and emotional support in times of need [54,55], has been suggested to predict psychological wellbeing [37,56,57]. Consistent evidence has pointed to an association between loneliness as correlating negatively with social support [58,59] and poorer physical and mental health outcomes including psychosis and depression [60,61,62,63]. 

In the context of an epidemic health crisis, such as the COVID-19 pandemic, research has indicated that social support plays an important role in providing a buffer against the negative psychological wellbeing outcomes that often emerge. Following the 2003 SARS outbreak in Hong Kong, social support buffered negative self-perceptions and had a beneficial impact on quality of life [64,65]. Likewise, in a review of Middle East Respiratory Syndrome (MERS), Ebola, and H5N1 outbreaks, seeking social support was correlated with greater life satisfaction [66]. When examining the COVID-19 literature, researchers have suggested that social support can be a powerful determinant of psychological wellbeing outcomes with studies showing perceived social support acting as a buffer between psychological health and COVID-19 worry [67] and a reduction of psychological distress through unneighborly support [68]. For LGBTQ+ individuals, social support was found to be of particular importance during the first wave of the COVID-19 pandemic; analyses conducted between March and July 2020 found that social support modulation effects between perceived stress and depression were four times greater for SGM individuals than their cisgender heterosexual peers [69]. Thus, the current study was designed to consider whether the protective effects of social support on psychological wellbeing would remain as the COVID-19 pandemic receded and associated social restrictions were lifted.

### 1.4. Categories of Social Support 

While much of the research cited above examined the overall influence of social support, delineating between categories may be helpful to understand how discrete areas of support interact differently with negative and positive psychological wellbeing outcomes. When reviewing the literature in this way, varied findings surface contrasting the effects of support from a special person (i.e., a significant other), family, or friends. The extant research field has revealed differences in wellbeing by category of social support in conjunction with SGM status. Furthermore, a handful of studies have pointed to these connections during the first wave of the COVID-19 pandemic in 2019–2020 when social restriction measures in the UK impacted the accessibility of different sources of social support. 

In their seminal paper, Cohen and Wills emphasized that the detrimental effects of stress on psychological health outcomes can be significantly diminished by being in a supportive couple relationship [52]. Likewise, community cohort studies from the first wave of the COVID-19 pandemic revealed that psychological wellbeing was generally better among people who lived with others rather than those who lived alone [70]. The importance of the social support nurtured within a close couple relationship also has been noted in many studies of LGBTQ+ adults, particularly in relation to support in managing minority stress [71]. Since Weston’s ground-breaking study [72], the importance of family of choice—intentional kin—for SGM people has been repeatedly emphasized in research showing that constructed families, made up of current and former partners and close friends, regularly provide the type of social support that otherwise might have been expected from biological relatives. However, studies tend not to differentiate among different sources of support from family of choice members, and few studies have considered the impact of partner support for SGM people specifically. Nonetheless, one study of lesbian, gay, and heterosexual married couples found that support from an intimate partner buffered wellbeing outcomes against stress, irrespective of sexual orientation [73], and a qualitative study of LGBT+ older people during the COVID-19 pandemic found being in a couple relationship to be predictive of better wellbeing [74]. 

Friendship has been identified as an important category of social support in buffering the relationship of stress and loneliness in young people [75]. For older adults, friendship may be important because of the types and frequency of activity with friends in providing an escape from everyday mundanities of home [76]. Furthermore, it has been suggested that friendship is of particular importance for older LGBT individuals, perhaps because they are less likely to report being in an intimate relationship than their cisgender heterosexual counterparts and may lack support from family members [77,78]. It may also be the case that family of choice networks have not yet been fully developed among many SGM young adults, particularly when the social spaces for meeting other LGBTQ+ youth are restricted [79] as these were in the UK and elsewhere during the pandemic [80]. 

Prior research studies have highlighted intergenerational (family) relationships as a particular source of social support promoting both physical and psychological wellbeing benefits across a person’s lifespan [81]. Family support has also been identified as a dominant determinant of psychological wellbeing for LGBTQ+ young people, with support from family members linked to lower levels of depression, PTSD, and psychological distress, whereas the absence of family support has been associated with higher levels of youth distress [82,83,84]. Nevertheless, in many families of origin, social support for an LGBTQ+ youth may be conditional upon conformity to normative expressions of gender and sexuality [85]. During the pandemic, studies have indicated that LGBTQ+ youth with the poorest psychological adjustment profiles were those who also lacked social support, particularly from family members [18,86]. Furthermore, during the first wave of the COVID-19 pandemic, some studies identified social support from family (not that from peers or a partner) to be the social support dimension most associated with lower levels of depression and post-traumatic stress among young adults [87]. Only 37% of LGBTQ+ youth rated their parental home as offering affirmation for their SGM identity [88]. Moreover, research conducted during the first wave of the COVID-19 pandemic found that psychological wellbeing tended to deteriorate among sexual minority adults who had returned to live in their parents’ house because of mandated restrictions on socializing between households [89]. 

### 1.5. Research Aims 

The current study explored the importance of different sources of social support—from a significant other or partner, family, or friends—on psychological wellbeing in the aftermath of the COVID-19 pandemic for both LGBTQ+ and cisgender heterosexual individuals living in the UK. With an increasing number of people identifying as LGB+ in the UK (3.2% in 2021) [90], understanding whether the psychosocial support needs of the LGBTQ+ community differ from those of their cisgender heterosexual contemporaries is of paramount importance. 

Would the detriment to LGBTQ+ psychological wellbeing found in many studies conducted during the first wave of the COVID-19 pandemic in 2020 continue in the aftermath of the pandemic as restrictions lifted in the UK in 2022? Based upon the current literature, we hypothesized that LGBTQ+ participants would report higher levels of negative psychological wellbeing outcomes (symptoms of depression, anxiety, and stress) and lower levels of positive wellbeing (life satisfaction) than their cisgender heterosexual peers. Secondly, we predicted that different sources of social support would be significantly associated with negative and positive psychological wellbeing outcomes and that differences between and LGBTQ+ and cisgender heterosexual groups would emerge. 

## 2. Materials and Methods

### 2.1. Participants

The sample comprised 1330 respondents. Almost half of the participants identified as LGBTQ+ (44.7%). Just over half the total sample identified as female (51.5%), 25.7% male, and 12.8% non-binary, with 10% undisclosed. The mean age was 44.94 years (SD = 17.16). 

### 2.2. Procedures 

Participant recruitment took place through a variety of community engagement avenues: sharing the survey link with LGBTQ+ networks, advertising on social media sites (e.g., Facebook and Instagram) and online forums (e.g., nextdoor.co.uk, reddit.co.uk), and placing physical flyers placed in public spaces, such as LGBTQ+ venues, supermarkets, and leisure centers. Data were collected between 24 March and 7 June 2022 via an online survey, which took approximately 25–30 min to complete. Upon opening the web link and before commencing the survey proper, all participants were provided with a briefing document outlining the purpose of the study, who could take part, what participation involved, and potential risks and benefits. Anonymity, confidentiality, and right to withdraw arrangements were outlined, and contact details were provided for comments, questions, or concerns. Informed consent was denoted by clicking to access the survey. Upon completion of the survey, a debriefing screen was presented.

### 2.3. Measures

#### 2.3.1. Sociodemographic and Personality Factors

The survey opened with questions and answer options designed to capture sociodemographic information including age, gender, sexual identity, disability, and ongoing physical or mental health illness based upon the UK Government Census-2021 questions [91]. In addition, two personality scales were included in the body of the survey to measure individual differences previously associated with the propensity to experience psychological stress, depression, and anxiety. Our measure of neuroticism was derived from the NEO Five-Factor Inventory (NEO-FFI), an abridged eight-item version of the original Big Five Inventory [92]. Participants scored each presented statement from 1 to 5 (disagree strongly to agree strongly). A revised version of the Life Orientation Test (LOT-R) was used as a measure of optimism; this comprised six statements enabling survey respondents to register a score between 0 and 4 (strongly disagree to strongly agree) on each item [93]. The distribution and mean differences between LGBTQ+ respondents and cisgender heterosexual respondents on sociodemographic and personality variables are displayed in Table 1.

Independent *t*-tests were run to investigate differences between LGBTQ+ and cisgender heterosexual groups for age, level of neuroticism (NEO-FFI) and level of optimism (LOT-R). A significant difference was found for age [t(1145.706) = 18.47, *p* < 0.001], level of neuroticism [t(1055.275) = −10.215, *p* < 0.001], as well as level of optimism [t(837) = 6.996, *p* < 0.001] with the LGBTQ+ group being significantly younger than the cisgender heterosexual group plus reporting higher levels of neuroticism and lower levels of optimism. In addition, chi-square tests were conducted to examine whether there were significant differences between the LGBTQ+ group versus the cisgender heterosexual group in the proportions of people reporting a pre-existing health condition as well as the number of individuals identifying as female versus male or nonbinary. A statistically significant association was found for health condition versus no health condition (x^2^ [N = 1141] = 71.37, *p* < 0.001) as well as for female versus the male and nonbinary categories combined (x^2^ [N = 1175] = 215.72, *p* < 0.001).

#### 2.3.2. Categories of Social Support

Perceived social support was measured using the Multidimensional Scale of Perceived Social Support (MSPSS) [94]. The MSPSS is a 12-item questionnaire with questions pertaining to three differing areas of social support: family, friend, and significant other (their partner or significant other, who could be their best friend). Participants were asked to rate each statement from 1 to 7 (very strongly disagree to very strongly agree). Cronbach alphas on the subscales measuring family, friends, and significant other were as follows: family α = 0.94, friends α = 0.94, and significant other α = 0.96.

#### 2.3.3. Psychological Wellbeing Variables

To assess negative psychological wellbeing, we deployed the Depression, Anxiety and Stress Scale (DASS-21) composed of three separate subscales measuring depressing thought patterns, anxiety thought patterns, and feelings of stress with each subscale measured using seven items [95,96]. Participants recorded a response to each item from 0 (did not apply to me at all) through to 3 (applied to me very much or most of the time). Participants were made aware that these questions were included to gauge wellbeing levels and were asked to answer based on feelings from the past week. In the analyses reported below, we use the DASS-21 total score (the sum of all 21 items on the scale) as the outcome measure of negative psychological wellbeing. In the current sample, the Cronbach’s alpha for the DASS-21 scale was α = 0.956.

To measure positive wellbeing, the survey included the Satisfaction With Life Scale (SWLS-3) consisting of three items with response options presented of between 1 and 7 (strongly disagree to strongly agree) for each item [97]. In the present study, the Cronbach’s alpha for the SWLS-3 scale was α = 0.91.

### 2.4. Statistical Analyses 

Data analyses were conducted using the IBM Statistical Package for the Social Sciences version 26 (SPSS (IBM Inc., Chicago, IL, USA)) [98]. We first conducted *t*-tests to compare negative and positive psychological wellbeing levels and social support evaluations of survey participants who identified as LGBTQ+ with those who identified as cisgender heterosexual.

Next, two parallel hierarchical multiple regression (HMR) analyses were conducted to establish the variables associated with negative and positive psychological wellbeing outcomes for cisgender heterosexual individuals and separately for individuals who identified as LGBTQ+. HMR predictor variables were grouped within three nested models in each HMR to examine whether social support explained variance in negative and positive psychological wellbeing reported by cisgender heterosexual and LGBTQ+ individuals over and above relevant sociodemographic and personality characteristics. In model 1, sociodemographic predictors of age, gender, and existing physical and mental health conditions were entered. To effectively consider gender in the HMR analyses for cisgender heterosexual and LGBTQ+ subsamples, we created two dummy variables for gender in the LGBTQ+ group containing n = 169 nonbinary participants. Thus, we considered women separately from a combined group of men and nonbinary people, and then men were considered separately from a combined group of women and nonbinary people.

Model 2 added personality data into the HMR, namely, the neuroticism and optimism scale scores. Finally, the three variables of interest were introduced in model 3: social support from family, friends, and a special person. Correlations between the independent variables were all <0.8 [99]. We used tolerance and VIF as multicollinearity indexes; the most common cut-off is a tolerance value >0.10 corresponding to a VIF < 10 [99]. All tolerance values were >0.10 and VIF values fell between 1 and 3.

Our dependent variables (DVs) deployed as individual outcome variables in the HMRs were the psychological wellbeing scores recorded using the total score from the DASS-21 inventory (negative psychological wellbeing) and the satisfaction with life total score from the SWLS-3 (positive psychological wellbeing). 

The survey sample size resulted in the following statistical power: for the independent *t*-tests, the power was 0.93, 1.00, and 1.00 for small, medium, and large effect sizes, respectively, and 0.30, 0.99, and 0.99 for small, medium, and large effect sizes, respectively, for the HMR with the smallest sample size (n = 171). 

To further refine our analyses, we conducted sensitivity analyses to establish whether adding two further variables related to the COVID-19 pandemic experience would make a difference to our HMR results [18]. Specifically, our survey included the question: How afraid are you of becoming infected with COVID-19 in the future? This question was answered on a scale ranging from 0 to 10 (not afraid at all to totally afraid). We also asked participants whether any member of their family (or a close friend) had been infected with COVID-19; responses were reported as ‘yes’ versus ‘I am unsure’ or ‘no’. Both pandemic experience variables were entered as a penultimate step in the HMRs conducted prior to entering social support from family, friends, and a special person in the last step of the model. We found that these COVID-19 pandemic experience variables accounted for only a very small proportion of the explained variables (R^2^ = 1%) for both negative or positive psychological wellbeing outcomes in either the LGBTQ+ or the cisgender heterosexual sample. Thus, we have reported our results without the inclusion of these extraneous variables.

## 3. Results

### 3.1. Cisgender Heterosexual and LGBTQ+ Differences in Psychological Wellbeing and Social Support Levels

Independent *t*-test analyses revealed significant differences between the cisgender heterosexual versus LGBTQ+ groups for DASS-21 total scores (measuring negative psychological wellbeing), with LGBTQ+ participants in general reporting a poorer outcome (see Table 2 below for subsample comparison statistics). In contrast, positive psychological wellbeing levels did not differ since both cisgender heterosexual and LGBTQ+ participant groups reported similar levels of general satisfaction with life. When reviewing differences in social support between cisgender heterosexual and LGBTQ+ groups, LGBTQ+ respondents perceived significantly lower levels of support from family and higher levels of support from friends than did the cisgender heterosexual group. However, no significant between group difference was apparent regarding support received from participant’s nominated special person. 

### 3.2. Hierarchical Multiple Regression Analysis: Exploring Associations of Social Support with Psychological Wellbeing 

To establish the impact of perceived social support on psychological wellbeing outcomes, six HMR models were conducted. Based upon nine predictor variables, the sample sizes for each model were satisfactory. A correlation matrix revealed significant correlations between variables (Table 3). Tests for independence of errors with the Durbin Watson calculation were non-significant. 

#### 3.2.1. Negative Psychological Wellbeing: Depression, Anxiety, and Stress Levels

When reviewing the HMR for cisgender heterosexual respondents in relation to negative psychological wellbeing, both age and existing health condition were significant predictors of the outcome in model 1, whereas gender was not (see Table 4). Age had a negative relationship suggesting that younger adults reported worse outcomes. Health condition had a positive association, indicating that those with a health condition (either a mental health or a physical condition or both) reported worse psychological wellbeing outcomes. At this stage, 14.9% of the variance was accounted for. In model 2, both age and health condition remained significant, as were both personality measures, namely, neuroticism and optimism. Neuroticism had a positive association with negative psychological wellbeing scores (those with higher levels of neuroticism had higher levels of depression, anxiety, and stress than did those with lower levels). Life orientation had a negative relationship with depression, anxiety, and stress scores; pessimists reported worse wellbeing outcomes than did those who had a more optimistic general approach to life. The variance at this stage increased to 46%. Model 3 was not significant, with no category of social support explaining a statistically significant amount of the variance in negative psychological wellbeing scores among our group of cisgender heterosexual participants. The total amount of variance in negative psychological wellbeing scores explained by the HMR models as a whole was 44.8%.

In the LGBTQ+ HMR model 1, which accounted for 24.5% of the explained variance in negative psychological wellbeing scores, neither existing health condition nor gender were significant, and only age added a significant proportion to the variance explained (see Table 5). Here, age had a negative relationship with negative psychological wellbeing recorded by LGBTQ+ participants, and this significance was retained in the subsequent HMR models run congruent with the pattern obtained in Table 4 above for the cisgender heterosexual subsample. However, in contrast to the HMR model 1 pattern for cisgender heterosexual participants, having an underlying health condition was not significantly associated with variation in negative wellbeing scores among LGBTQ+ participants. In model 2, higher levels of neuroticism were significantly associated with higher scores, indicating significant levels of depression, anxiety, and stress. Life orientation likewise displayed a similar pattern of association with negative wellbeing scores among our LGBTQ+ participants as it did for those in the cisgender heterosexual group. Repeating the pattern seen for our cisgender heterosexual participants, model 3 was not statistically significant for LGBTQ+ participants either; no category of social support explained a significant amount of variance in negative psychological wellbeing scores. In total, 52.4% of the variance in negative wellbeing scores was explained by the set of HMRs with LGBTQ+ participants’ data. 

#### 3.2.2. Positive Psychological Wellbeing: Satisfaction with Life Scores 

Analysis of the HMR model for positive wellbeing scores given by cisgender heterosexual individuals revealed that having an existing health condition had a significant negative association with SWLS-3 scores: those with no physical or mental health condition in HMR model 1 reported higher levels of life satisfaction (see Table 6). Neither age nor gender were significant, with 9.1% variance accounted for. Having an underlying health condition retained statistical significance in model 2, and the amount of explained variance in satisfaction with life scores was boosted with the addition of neuroticism and life orientation scores (with an additional 28.8% of the variance explained in model 2). Neuroticism had a negative relationship, demonstrating that individuals with higher neuroticism scores reported lower satisfaction with life levels, whereas life orientation had a positive relationship, with optimists reporting being more satisfied with their lives in comparison to pessimists. In the final HMR, having an underlying health condition, together with scores on the neuroticism and life orientation inventories, all retained statistical significance in relation to positive wellbeing scores. Further, in model 3, the addition of the social support subscale scores increased the total variance explained to 44.7% of the variation in cisgender heterosexual participants’ satisfaction with life scores. Both support from family and support from a special person showed significant positive relationships with positive wellbeing, indicating that those who reported higher levels of perceived support from either group had higher levels of life satisfaction. Support from friends made no significant contribution to the model.

In the equivalent HMR for LGBTQ+ participants, the variance explained by model 1 was 11.9% (see Table 7). Similar to the HMR results with the cisgender heterosexual subsample, LGBTQ+ participants declaring an existing health condition also reported lower satisfaction with life, but neither age nor gender significantly contributed to life satisfaction scores. In model 2, health condition retained statistical significance, and the explained variance in life satisfaction scores of LGBTQ+ participants increased to 31.4%. As with cisgender heterosexual participants, life orientation also accounted for a significant proportion of the explained variance in our LGBTQ+ group; those who were more optimistic were also more satisfied with life. Among our LGBTQ+ participants, neuroticism scores did not make a significant contribution to explaining life satisfaction scores, in contrast to the significant pattern of results within the cisgender heterosexual group. In the final HMR model for our LGBTQ+ subsample, the overall variance in positive wellbeing accounted for 39.4%. For LGBTQ+ participants, their life orientation approach and having an underlying health condition both retained statistical significance, and higher scores on social support from family and support from a nominated special person also significantly contributed to the proportion of positive wellbeing variance explained. Again, as noted for the cisgender heterosexual subsample, the extent of social support received from friends was not associated with the life satisfaction scores of our LGBTQ+ participants (Table 7). 

## 4. Discussion

The present study examined whether different sources of social support—from a partner or special person, family members, or friends—were associated with differences in psychological wellbeing levels among LGBTQ+ and cisgender heterosexual individuals. Our online survey was completed by participants living in the UK two years after the start of the COVID-19 pandemic, just as the UK government announced the final easing of social distancing restrictions in the community. Our first set of hypotheses derived from previous research conducted during the first wave of the COVID-19 pandemic [16,17,25] were partially supported by our current findings as restrictions lifted. Specifically, overall higher levels of depression, anxiety, and stress symptoms were recorded by participants in our LGBTQ+ subsample than those in our cisgender heterosexual group. However, in the wake of the pandemic, no group differences in positive wellbeing were found between LGBTQ+ and cisgender heterosexual participants in the current study. Prior research specifically focused on LGBTQ+ satisfaction with life ratings have produced mixed results. On the one hand, some studies have indicated that sexual minority people report a reduced quality of life [26,28]. On the other hand, findings from other studies reveal a more nuanced set of results. Gay and bisexual men reported lower life satisfaction than did heterosexual men as did bisexual women in both the UK and Australia, but no significant differences in life satisfaction rates were found between UK lesbian and heterosexual women despite a significant reduction in scores among lesbian women in Australia [27]. Perhaps our own findings of no difference in life satisfaction ratings might reflect the preponderance of women in the UK sample surveyed. 

Our second set of hypotheses that social support would be significantly associated with psychological wellbeing outcomes was again partially supported. In the current study no particular source of social support was associated with variation in depression, anxiety, or stress scores. Here, results from the present study can be contrasted with findings from previous studies conducted during the earlier phases of the pandemic in which increased social support was found to be associated with greater psychological wellbeing [67]. For SGM individuals specifically, Jacmin-Park and colleagues found social support to be 400% more effective in buffering connections between stress and depression in comparison to findings recorded among their cisgender heterosexual peers [69]. The results of the present study revealed no such association and no greater impact for LGBTQ+ than cisgender heterosexual people. The contrast in findings may be accounted for by the later timing of the current survey when social restrictions ceased to be in operation; thus, it is possible that social support was at a lower premium than it had been earlier in the pandemic when in-person support was restricted to household contacts for most people.

Notwithstanding, our findings do highlight the role of social support in relation to positive psychological wellbeing. Support from family and/or a special person had a significant association with satisfaction with life; those recording greater social support from either source experienced higher life satisfaction levels. This was true for both LGBTQ+ and cisgender heterosexual individuals; however, the strength of associations was marginally greater for the LGBTQ+ group regarding the importance of support from a nominated special person. In neither set of HMRs for LGBTQ+ or cisgender heterosexual participants was friendship a significant predictor of how satisfied individuals were with their lives. 

To date, researchers have paid far less attention to the factors associated with positive psychological wellbeing compared to negative wellbeing. Prior studies have pointed to the importance of social support for satisfaction with life levels [64] and in particular during a disease outbreak crisis [66]. Our study has expanded the field by noting that support from family or support from a partner or special person has significance over and above the general level of social support provided by friends and others.

Aside from levels of social support recorded, the following factors are also associated with psychological wellbeing; having an underlying health condition, neuroticism, and life orientation approach all made a significant difference to how satisfied an individual was with their life. Namely, younger people, those with higher levels of neuroticism, and those with lower levels of optimism reported increased levels of depression, anxiety, and stress. Here, results from the present study were in line with a previous paper which found that individuals under 35 years of age have the highest levels of mental health problems [40]. Furthermore, research conducted during the recent pandemic found that neuroticism significantly predicted depression and anxiety levels [46] and that optimism was associated with lower levels of anxiety and depression [47]. Again, as noted in earlier studies, having a pre-existing health condition was found to be predictive of higher reported levels of depression and anxiety symptoms [7,50,51]. However, in the present study that was conducted as pandemic-related social contact restrictions lifted, only within our subsample of cisgender heterosexual individuals did having an existing health condition make a difference to reported levels of depression, anxiety, and stress symptoms. Our cisgender heterosexual subsample was composed of proportionately more women than men, who were on average older and less likely than our LGBTQ+ subsample to report having an underlying physical and/or mental health condition; thus, any health threats may perhaps have raised more noticeable existential concerns among this group of respondents.

### Limitations

Our survey was cross-sectional and not longitudinal; thus, the data revealed only associations between participant characteristics or social support and psychological wellbeing, not potential pathways. Whilst over 1300 individuals responded to the survey, when it came to questions relevant to the outcome variables, sample sizes diminished. As the survey was designed to capture a wealth of information above and beyond that which has been explored in this paper, respondent fatigue may have led to unanswered questions, thus producing smaller sample sizes. 

Nevertheless, we have situated our findings alongside those from pre-pandemic studies and from prior studies conducted during the early waves of the COVID-19 pandemic. In addition, the results from our research sit within the context of other studies that have considered the psychological wellbeing of LGBTQ+ groups during the pandemic. To date, few studies have considered potential psychological wellbeing distinctions between different sexual and gender minority groups. Previous studies have reported that binary transgender and non-binary people have a higher risk of depression, anxiety, and suicide and experience greater challenges accessing mental health services than did cisgender heterosexual individuals and that transgender people have been disproportionally affected by the pandemic [15,100,101]. Therefore, a potential further direction of research may be to focus the data analysis on these groups specifically to understand how experiences vary not only from cisgender heterosexual peers but also within the wider LGBTQ+ community grouping and whether the impact of social support depends upon specific sexual or gender minority status. 

Given that we found associations between social support and positive psychological wellbeing, we recommend that future investigations pay particular attention to the various components of positive wellbeing, such as spiritual wellbeing and the assessment of subjective wellbeing [102].

## 5. Conclusions

Our study considered the psychological wellbeing of LGBTQ+ and cisgender heterosexual individuals two years on from the beginning of the COVID-19 pandemic, just as the restrictions on social distancing were lifting in the UK. The results clearly point toward continued poorer psychological wellbeing among LGBTQ+ individuals than among cisgender heterosexual people. Neither social support from a partner (or nominated special person), family members, nor friends made a significant difference to the reported level of depression, anxiety, and stress for either LGBTQ+ or cisgender heterosexual groups. In contrast, different sources of social support—from a special person or family members—boosted positive psychological wellbeing levels for both LGBTQ+ and cisgender heterosexual individuals, and the rates of positive psychological wellbeing levels were equivalent across both subsamples in terms of their general satisfaction with life two years after the pandemic was declared. Our findings highlight the need to consider positive wellbeing at times of crisis and how social support can play a key role in improving satisfaction with life. 

## Figures and Tables

**Table 1 healthcare-12-01634-t001:** Distribution and means for LGBTQ+ and cisgender heterosexual respondents for sociodemographic and personality variables.

Variable	LGBTQ+ (n = 595)		Cisgender Heterosexual (n = 594)	
	M	SD	M	SD
Age	36.77	14.238	55.11	15.918
Neuroticism	27.55	6.354	23.35	7.103
Optimism	10.78	5.007	13.17	4.863
	n	%	n	%
Female	209	35.1	460	77.4
Male	208	35	129	21.7
Non-Binary	169	28.4	0	0
Health Condition	362	60.8	224	37.7

**Table 2 healthcare-12-01634-t002:** Results comparing cisgender heterosexual and LGBTQ+ participants on psychological wellbeing and types of social support.

Variable	Cisgender-Heterosexual			LGBTQ+			t	*p*	Cohen’s *d*	95% CI for Cohen’s *d*	
	n	M	SD	n	M	SD				Lower	Upper
DASS-21	318	13.92	12.76	189	21.39	16.92	−5.25	<0.001	−0.52	−0.70	−0.33
SWLS	420	12.62	4.90	430	11.06	4.69	4.76	0.495	0.33	0.19	0.46
Family support	443	19.23	6.30	437	16.3	6.72	6.66	0.042	0.45	0.32	0.58
Friend support	441	19.06	5.77	435	29.89	4.97	−5.05	0.008	−0.34	−0.47	−0.21
Special person support	445	21.14	6.62	438	21.68	6.44	−1.24	0.992	−0.08	−0.22	0.05

**Table 3 healthcare-12-01634-t003:** Table of correlations for all variables.

	Age	Woman	Man	LGBTQ+	Health Condition	Neuroticism	Optimism	Family Support	Friend Support	SP Support	DASS Total	SWLS
Age		0.233 **	0.000	−0.476 **	−0.105 **	−0.337 **	0.258 **	0.087 **	−0.102 **	−0.032	−0.350 **	0.152 **
Woman ^a^			−0.732 **	−0.427 **	−0.096 **	−0.041	0.147 **	0.182 **	0.054	0.051	−0.061	0.099 **
Man ^b^				0.151 **	−0.069 **	−0.117 **	0.006	−0.079 *	−0.106 **	−0.078 *	−0.077	−0.008
LGBTQ+ ^c^					0.251 **	0.298 **	−0.235 **	−0.219 **	0.168 **	0.042	0.243 **	−0.161 **
Health Condition ^d^						0.315 **	−0.276 **	−0.225 **	−0.064	−0.038	0.322 **	−0.329 **
Neuroticism							−0.595 **	−0.202 **	−0.143 **	−0.065	0.650 **	−0.427 **
Optimism								0.260 **	0.270 **	0.181 **	−0.590 **	0.551 **
Family Support									0.328 **	0.430 **	−0.206 **	0.379 **
Friend Support										0.509 **	−0.221 **	0.308 **
SP Support											−0.178 **	0.366 **
DASS-21 Total												−0.562 **
SWLS												

Note: ^a^ Woman: 0 = other, 1 = female; ^b^ Man: 0 = other, 1 = male; ^c^ LGBTQ+: 0 = cishet, 1 = LGBTQ+; ^d^ Health condition: 0 = good health, 1 = poor health; * *p* < 0.05; ** *p* < 0.01.

**Table 4 healthcare-12-01634-t004:** Summary of hierarchical regression analysis for total depression, anxiety, and stress scores (DASS-21) for cisgender heterosexual participants (n = 284).

Variable	Model 1			Model 2			Model 3		
	B	SEB	β	B	SEB	β	B	SEB	β
Age	−0.172	0.047	−0.204 ***	−0.087	0.038	−0.103 *	−0.098	0.038	−0.117 *
Woman ^a^	−2.274	11.965	−0.075	−6.543	9.577	−0.215	−7.279	9.546	−0.239
Man ^b^	−3.509	12.012	−0.115	−6.333	9.627	−0.207	−7.416	9.631	−0.243
Health condition ^c^	8.519	1.446	0.327 ***	3.756	1.222	0.144 **	4.099	1.221	0.157 **
Neuroticism				0.741	0.104	0.415 ***	0.724	0.105	0.405 ***
Optimism				−0.673	0.144	−0.257 ***	−0.596	0.147	−0.227 ***
Family support							0.197	0.118	0.094
Friend support							−0.195	0.129	−0.087
Special person support							−0.170	0.115	−0.086
*R* ^2^		0.149			0.460			0.472	
*F* for change in *R*^2^		12.208 ***			79.607 ***			2.233	
*R*^2^ change		0.149			0.311			0.013	
Adjusted *R*^2^		0.137			0.448			0.455	

Note. ^a^ Woman: 0 = male, 1 = female; ^b^ Man: 0 = female, 1 = male; ^c^ Health cond: 0 = good health, 1 = poor health; * *p* < 0.05; ** *p* < 0.01; *** *p* < 0.001.

**Table 5 healthcare-12-01634-t005:** Summary of hierarchical regression analysis for total depression, anxiety, and stress scores (DASS-21) for LGBTQ+ participants (n = 171).

Variable	Model 1			Model 2			Model 3		
	B	SEB	β	B	SEB	β	B	SEB	β
Age	−0.476	0.085	−0.397 ***	−0.173	0.073	−0.147 *	−0.158	0.073	−0.135 *
Woman ^a^	−5.275	2.944	−0.155	−2.799	2.322	−0.082	−2.968	2.320	−0.87
Man ^b^	−6.073	3.070	−0.182	−4.003	2.426	−0.120	−4.796	2.457	−0.144
Health condition ^c^	4.540	2.329	0.135	1.890	1.848	0.056	1.469	1.939	0.044
Neuroticism				0.726	0.166	0.317 ***	0.728	0.168	0.318 ***
Optimism				−1.072	0.220	−0.357 ***	−0.968	0.228	−0.322 ***
Family support							0.009	0.142	0.004
Friend support							−0.248	0.192	−0.081
Special person support							−0.113	0.145	−0.049
*R* ^2^		0.245			0.541			0.552	
*F* for change in *R*^2^		13.446 ***			52.983 ***			1.241	
*R*^2^ change		0.245			0.296			0.010	
Adjusted *R*^2^		0.227			0.524			0.526	

Note. ^a^ Woman: 0 = male/non-binary, 1 = female; ^b^ Man: 0 = female/non-binary, 1 = male; ^c^ Health cond: 0 = good health, 1 = poor health. * *p* < 0.05; *** *p* < 0.001.

**Table 6 healthcare-12-01634-t006:** Summary of hierarchical regression analysis for Satisfaction With Life Scale (SWLS-3) scores for cisgender heterosexual participants (n = 373).

Variable	Model 1			Model 2			Model 3		
	B	SEB	β	B	SEB	β	B	SEB	β
Age	0.011	0.015	0.037	−0.015	0.013	−0.051	−0.007	0.012	−0.025
Woman ^a^	−1.040	4.649	−0.089	0.785	3.857	0.067	2.120	3.660	0.182
Man ^b^	−1.418	4.664	−0.121	0.872	3.873	0.075	2.577	3.684	0.220
Health condition ^c^	−2.945	0.490	−0.300 ***	−1.622	0.428	−0.165 ***	−1.598	0.407	−0.163 ***
Neuroticism				−0.094	0.036	−0.141 **	−0.087	0.035	−0.130 *
Optimism				0.464	0.050	0.471 ***	0.405	0.050	0.411 ***
Family support							0.089	0.040	0.115 *
Friend support							0.020	0.044	0.024
Special person support							0.125	0.039	0.172 **
*R* ^2^		0.091			0.378			0.447	
*F* for change in *R*^2^		9.167 ***			84.685 ***			15.049 ***	
*R*^2^ change		0.091			0.288			0.069	
Adjusted *R*^2^		0.081			0.368			0.433	

Note. ^a^ Woman: 0 = male, 1 = female; ^b^ Man: 0 = female, 1 = male; ^c^ Health cond: 0 = good health, 1 = poor health; * *p* < 0.05; ** *p* < 0.01; *** *p* < 0.001.

**Table 7 healthcare-12-01634-t007:** Summary of hierarchical regression analysis for Satisfaction With Life Scale (SWLS-3) scores for LGBTQ+ participants (n = 402).

Variable	Model 1			Model 2			Model 3		
	B	SEB	β	B	SEB	β	B	SEB	β
Age	0.058	0.017	0.171 **	0.018	0.015	0.054	0.017	0.015	0.050
Woman ^a^	0.526	0.543	0.055	0.117	0.482	0.012	0.030	0.456	0.003
Man ^b^	−0.149	0.583	−0.015	−0.634	0.519	−0.065	−0.270	0.495	−0.028
Health condition ^c^	−2.781	0.465	−0.289 ***	−1.606	0.428	−0.167 ***	−1.359	0.416	−0.141 **
Neuroticism				−0.071	0.039	−0.097	−0.081	0.037	−0.111
Optimism				0.381	0.049	0.418 ***	0.304	0.047	0.333 ***
Family support							0.080	0.029	0.117 **
Friend support							0.038	0.042	0.040
Special person support							0.161	0.033	0.224 ***
*R* ^2^		0.119			0.314			0.394	
*F* for change in *R*^2^		13.348 ***			56.371 ***			17.218 ***	
*R*^2^ change		0.119			0.196			0.080	
Adjusted *R*^2^		0.110			0.304			0.380	

Note. ^a^ Woman: 0 = male/non-binary, 1 = female; ^b^ Man: 0 = female/non-binary, 1 = male; ^c^ Health cond: 0 = good health, 1 = poor health. ** *p* < 0.01; *** *p* < 0.001.

## Data Availability

The dataset presented in this article is not readily available due to time limitations. Requests to access the datasets should be directed to f.tasker@bbk.ac.uk.

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
