# Peer review of "Surveying Psychological Wellbeing in a Post-Pandemic World: The Role of Family and Social Support for LGBTQ+ and Cisgender Heterosexual Adults in the UK"

_healthcare, 2024, doi:10.3390/healthcare12161634_

Round 1
Reviewer 1 Report
Comments and Suggestions for Authors
This paper evaluates the role of social support in mitigating the relationship between psychological distress and mental health effects during the time when the COVID-19 pandemic restrictions lifted. The paper is very well written, the science well-explained and documented, and the study design simple but effective. A few minor concerns remain for the authors’ consideration. Details included below:
· Missing word: “Family support also been identified as a dominant determinant of psychological wellbeing for LGBTQ+ young people, with support from family members linked to lower levels of depression, PTSD, and psychological distress, whereas the absence of family support has been associated with higher levels of youth distress [82-84].”
· Is this missing the rest of the acronym? “LGB+”
· I believe this was introduced suddenly and without clear definition (although it is fairly easy to figure out). I would make a clear statement though introducing the term: “Cishet”
· Was recruitment done by matching? How is it almost the exact sample size for LGBTQ+ and cisgender heterosexual?
· Be sure to fully write out hierarchical multiple regression analysis the first time it is introduced.
· Instead of listing the variable for the results, can it be referred to as the construct? So not DASS-21 but negative psychological wellbeing.
· Which analyses are these? Be specific at the beginning for clarity. “Analyses revealed significant differences between cisgender-heterosexual and 288 LGBTQ+ groups for DASS-21 total scores, with LGBTQ+ participants in general reporting 289 a poorer outcome (see Table 2 below for subsample comparison statistics).”
· Revise Table 3 for APA formatting; also blurry in its current form.
· Check APA formatting for all tables, specifically number of decimal places and leading zeroes.
· This phrasing feels new (specifically the moderating language) if this is the goal, it should be clearer in the intro/data analysis. “The present study examined whether different sources of social support – from a partner or special person, family members, or friends – moderated differences in psycho- logical wellbeing levels among LGBTQ+ and cisgender-heterosexual individuals.”
Author Response
Comment 1: Missing word: “Family support also been identified as a dominant determinant of psychological wellbeing for LGBTQ+ young people, with support from family members linked to lower levels of depression, PTSD, and psychological distress, whereas the absence of family support has been associated with higher levels of youth distress [82-84].”
Response 1: This has been amended in the ms and now reads: “Family support has also been identified as a dominant determinant of psychological wellbeing for LGBTQ+ young people, with support from family members linked to lower levels of depression, PTSD, and psychological distress, whereas the absence of family support has been associated with higher levels of youth distress [82-84].” (See line 160).
Comment 2: Is this missing the rest of the acronym? “LGB+”
Response 2: Author’s reply: This acronym was used consciously rather than LGBTQ+, as it reflects the acronym used by the Office of National Statistics (see line 181 / reference 90).
Comment 3: I believe this was introduced suddenly and without clear definition (although it is fairly easy to figure out). I would make a clear statement though introducing the term: “Cishet”
Response 3: We agree with the reviewer and have removed this mention of “cishet” from the ms and replaced it with cisgender-heterosexual, inline with the rest of the paper.
Comment 4: Was recruitment done by matching? How is it almost the exact sample size for LGBTQ+ and cisgender heterosexual?
Response 4: Recruitment was not done by matching. To ensure as close to equal representation as possible, the researchers reached out to LGBTQ+ community organizations across the UK alongside advertising the survey to participants generally (e.g. via Facebook and other social media).
Comment 5: Be sure to fully write out hierarchical multiple regression analysis the first time it is introduced.
Response 5: We agree with the reviewer and this has been amended in the ms (see line 272).
Comment 6: Instead of listing the variable for the results, can it be referred to as the construct? So not DASS-21 but negative psychological wellbeing.
Response 6: We agree with the reviewer, and after the first mention in the abovementioned section, we have added the words (measuring negative psychological wellbeing), and thereafter replace DASS-21 in the body of the text, with negative psychological wellbeing). (See lines 316-317)
Comment 7: Which analyses are these? Be specific at the beginning for clarity. “Analyses revealed significant differences between cisgender-heterosexual and 288 LGBTQ+ groups for DASS-21 total scores, with LGBTQ+ participants in general reporting 289 a poorer outcome (see Table 2 below for subsample comparison statistics).”
Response 7: We agree with the reviewer and have added: ‘Independent t-test analyses, for clarity (see line 315)
Comment 8: Revise Table 3 for APA formatting; also blurry in its current form.
Response 8: This has been amended in the ms.
Comment 9: Check APA formatting for all tables, specifically number of decimal places and leading zeroes.
Response 9: This has been done in line with APA guidelines.
Comment 10: This phrasing feels new (specifically the moderating language) if this is the goal, it should be clearer in the intro/data analysis. “The present study examined whether different sources of social support – from a partner or special person, family members, or friends – moderated differences in psycho- logical wellbeing levels among LGBTQ+ and cisgender-heterosexual individuals.”
Response 10: We have amended the phrasing to: “The present study examined whether different sources of social support – from a partner or special person, family members, or friends – were associated with differences in psycho- logical wellbeing levels among LGBTQ+ and cisgender-heterosexual individuals.” (see lines 455-458).
Reviewer 2 Report
Comments and Suggestions for Authors
These are my comments
· The title should have the study design
· The abstract should be rich in numbers and their significance
· Please add the Cronback’s alpha to the (DASS-21) scale and the (MSPSS) scales
· Although the sample size is satisfactory, a sample size calculation is important to be added
· The statistics are well written and executed
· The tables should include in the legend the statistical analysis performed
Author Response
Comments 1: The abstract should be rich in numbers and their significance
Response 1: We have followed APA-7 style and usually APA abstracts don’t report statistics.
Comments 2: Please add the Cronback’s alpha to the (DASS-21) scale and the (MSPSS) scales
Response 2: We have calculated Cronbach’s alphas for the MSPSS subscales and our two dependent variables DASS-21 and the Satisfaction With Life Scale (SWLS). We have included information on these at the end of the subsections where we introduce these measures, namely, ‘2.3.2. Social Support’ and subsection ‘2.3.3. Psychological Wellbeing Variables'.
Comments 3: The tables should include in the legend the statistical analysis performed
Response 3: We have included the details of the statistical analyses performed in the text associated with each table – APA guidelines recommend avoiding redundant duplication in both the text and the table.
Caulfield, J. (2024, January 17). APA Format for Tables and Figures | Annotated Examples. Scribbr. Retrieved August 5, 2024, from https://www.scribbr.com/apa-style/tables-and-figures/
All other comments addressed and amended within the manuscript.
Reviewer 3 Report
Comments and Suggestions for Authors
I thank the authors for writing such an interesting and important manuscript. First, it is very well written, and therefore it is a joy to read. Research comparing LGBTQ+ adults and Cisgender heterosexual adults are not frequent, and therefore, this research brings much needed information.
The introduction I find is most excellent. The description of Meyer's minority stress model is very relevant and nuanced, with the authors presenting both versions of the model, the original from 2003, and its latest version from 2023.
Excellent nuance between social support and perceived social support and its impact on psychological well-being. Incidentally, the authors refer to 'psychological well-being' but there is also 'subjective well-being', from a positive psychology lens. I wonder if they might want to include that nuance, especially as the results show high levels of life satisfaction (associated with social support from family or special person) and that life satisfaction is a factor of subjective well-being.
So relevant that the authors would refer to other health crises and their impact on well-being.
'Family of choice' is so very important, and would in and of itself deserve an entire manuscript. That being said it is well described here and most relevantly included in the study.
Have the authors considered spiritual support and thus spiritual well-being? If not perhaps in another study?
Regarding the questionnaires: please include information on internal and external validity as well as reliability.
The acronym HRM is used for the first time on page 6 but it is spelled out (hierarchical multiple regressions models) on page 7. Please make sure to spell it out the very first time HRM is used. The acronym can be used hereafter.
Author Response
Comments 1: Excellent nuance between social support and perceived social support and its impact on psychological well-being. Incidentally, the authors refer to 'psychological well-being' but there is also 'subjective well-being', from a positive psychology lens. I wonder if they might want to include that nuance, especially as the results show high levels of life satisfaction (associated with social support from family or special person) and that life satisfaction is a factor of subjective well-being.
Response 1: Thank you for this interesting point. In the present investigation we used Diener’s Satisfaction With Life Scale and associated theory as we describe in our introduction and method section. We did not examine other aspects in our study, therefore we have decided not to include further background on this in the introduction. However, we agree that given our findings further distinctions in subjective wellbeing would be worthy topic for future investigations. We have included a sentence to this effect with a reference at the end of section 4.1 Limitations.
Comments 2: Have the authors considered spiritual support and thus spiritual well-being? If not perhaps in another study?
Response 2: We regrettably did not consider spiritual support or wellbeing in this sense. We have included a recommendation to measure this in future in the added sentence at the end of section 4.1 Limitations.
Comments 3: Regarding the questionnaires: please include information on internal and external validity as well as reliability.
Response 3: We have included Cronbach’s alphas as this was also suggested by Reviewer 2 (point 3). The originators of these standard scales give details of the internal and external validity of these tests and we have referred the reader to the relevant papers when we introduce the tests (e.g. citations 92-97).
Additional comments reviewed and amended within the manuscript.